# Lipid Metabolism Regulators Are the Possible Determinant for Characteristics of Myopic Human Scleral Stroma Fibroblasts (HSSFs)

**DOI:** 10.3390/ijms25010501

**Published:** 2023-12-29

**Authors:** Hiroshi Ohguro, Araya Umetsu, Tatsuya Sato, Masato Furuhashi, Megumi Watanabe

**Affiliations:** 1Department of Ophthalmology, School of Medicine, Sapporo Medical University, Sapporo 060-8556, Hokkaido, Japan; ooguro@sapmed.ac.jp (H.O.); araya.umetsu@sapmed.ac.jp (A.U.); 2Department of Cardiovascular, Renal and Metabolic Medicine, School of Medicine, Sapporo Medical University, Sapporo 060-8556, Hokkaido, Japan; 3Department of Cellular Physiology and Signal Transduction, School of Medicine, Sapporo Medical University, Sapporo 060-8556, Hokkaido, Japan

**Keywords:** sterol regulatory element-binding transcription factor (SREBF), myopia, human scleral stroma fibroblasts (HSSFs), RNA sequencing, cholesterol biosynthesis, insulin-induced gene 1 (INSIG1), insulin-like growth factor 1 (IGF1), protein tyrosine phosphatase receptor type O (PTPRO)

## Abstract

The purpose of the current investigation was to elucidate what kinds of responsible mechanisms induce elongation of the sclera in myopic eyes. To do this, two-dimensional (2D) cultures of human scleral stromal fibroblasts (HSSFs) obtained from eyes with two different axial length (AL) groups, <26 mm (low AL group, *n* = 2) and >27 mm (high AL group, *n* = 3), were subjected to (1) measurements of Seahorse mitochondrial and glycolytic indices to evaluate biological aspects and (2) analysis by RNA sequencing. Extracellular flux analysis revealed that metabolic indices related to mitochondrial and glycolytic functions were higher in the low AL group than in the high AL group, suggesting that metabolic activities of HSSF cells are different depending the degree of AL. Based upon RNA sequencing of these low and high AL groups, the bioinformatic analyses using gene ontology (GO) enrichment analysis and ingenuity pathway analysis (IPA) of differentially expressed genes (DEGs) identified that sterol regulatory element-binding transcription factor 2 (SREBF2) is both a possible upstream regulator and a causal network regulator. Furthermore, SREBF1, insulin-induced gene 1 (INSIG1), and insulin-like growth factor 1 (IGF1) were detected as upstream regulators, and protein tyrosine phosphatase receptor type O (PTPRO) was detected as a causal network regulator. Since those possible regulators were all pivotally involved in lipid metabolisms including fatty acid (FA), triglyceride (TG) and cholesterol (Chol) biosynthesis, the findings reported here indicate that FA, TG and Chol biosynthesis regulation may be responsible mechanisms inducing AL elongation via HSSF.

## 1. Introduction

Myopia is the most common refractive error leading to reversible visual impairment as well as blindness as the worst case worldwide [1,2,3]. In terms of the severity of myopia development, axial elongation of the ocular globe caused by unknown mechanisms exerts biomechanical stretching in the posterior segments of eyes, resulting in typical complications in the fundus such as posterior staphyloma, myopic maculopathy and diffuse choroidal atrophy [4,5].

The sclera, which is mainly composed of several extracellular matrix molecules (ECMs) including collagens (COLs), is well known as the barrier to protect the intra-ocular structures and as the critical determinant of the shape and size of the eye globe [6]. Alternatively, the sclera is also recognized as the most important pathological factor of myopia [7,8,9]. Among the structural components of the human sclera, human scleral stroma fibroblasts (HSSFs) producing and secreting ECMs are known to be involved in pivotal roles during the development of the eye [7,10,11,12]. HSSFs increase the production of ECMs productions until the globe reaches adult size at approximately 10 years of age [13], and then the production of ECMs is decelerated, and thereafter, the scleral ECM remodeling is maintained at physiological levels throughout the course of existence of an individual [14]. Although the pathogenesis of myopia has not been elucidated, it is speculated that some unidentified mechanisms in HSSF-related metabolism may lead to scleral elongation that is particularly evident in the posterior segment to form posterior staphyloma [15]. In fact, previous histopathological studies indicated that smaller in diameter and abnormally arranged COL fibrils [15] and downregulated ECM production are recognized in human and primate myopic sclera [16,17]. It was also demonstrated that the mRNA expression level of COL1 was significantly decreased compared to the expression levels of COL3 and COL5, suggesting an increase of more small-diameter COL fibers in the myopic myopia model [11]. Furthermore, it was shown that activated matrix metalloproteinases (MMPs) and decreased tissue inhibitor of metalloproteinases (TIMPs) enzymatic activities are involved in the progressing course of myopia-associated elongation of the sclera resulting in larger axial length (AL) [18,19]. However, at present, to elucidate the unknown responsible mechanism inducing such myopia-associated pathological changes, especially changes related to HSSFs, no suitable in vitro models replicating myopic pathogenesis have been available. In our preceding study, to establish a desirable in vitro model of myopia, a three-dimensional (3D) culture method [20,21,22,23] was employed using HSSFs obtained from seven patients, and these spheroids were categorized into three groups by axial length (AL): (1) less than 25 mm (*n* = 2), (2) 25–30 mm (*n* = 2) and (3) more than 30 mm (*n* = 3). Notably, the 3D HSSF spheroids originating from the more than >30 mm AL group were substantially down-sized and softened as compared to those in the other groups. In addition, there were significant alterations in the mRNA and protein expressions of (1) some major ECM proteins (up-regulated *COL1* and *COL4* and down-regulated *FN*), (2) ECM modulators (up-regulated MMP2 and MMP14 and down-regulated TIMP 2 and TIMP, (3) *HIF2A* (up-regulation), and (4) several ER stress-related genes (up-regulation of *XBP1* and down-regulation of *GRP78*, *GRP94*, *IRE1* and *ATF6*) with increasing AL length [24]. Based upon these collective observations, we suggested that HSSFs themselves could have different biological properties among the different AL groups.

In the current study, to obtain additional insights into the possible molecular mechanisms causing myopic changes in the sclera, HSSF cells obtained from two different AL groups, <26 mm (*n* = 2) and >27 mm (*n* = 3), were subjected to real-time cellular metabolism, mitochondrial and glycolysis, analyses to characterize biological functions as well as RNA sequencing analysis to identify the possible responsible molecules involved in the etiology of the scleral elongation in myopic eyes.

## 2. Results

First, extracellular flux analysis was performed to determine the cellular functions of HSSF cells obtained from five eyes with different ALs (Table 1). As shown in Figure 1, both OCR indices and ECAR indices as mitochondrial and glycolytic functions, respectively, were significantly diverse among them. That is, (1) both OCR and ECAR indices were highest in N1 (the shortest AL) and (2) ECAR indices were higher in N2 than in M1 through M3, suggesting decreased levels of cellular metabolic functions of HSSFs in myopic eyes. 

To identify the possible regulatory mechanisms responsible for inducing myopic scleral elongation, RNA sequencing analyses were performed using HSSF cells obtained from two AL groups, less than 26 mm (low AL, *n* = 2, N1: 22.84 mm, and N2: 25.94) and more than 27 mm (high AL, *n* = 3, M1: 27.27 mm, M2: 27.50 mm and M3: 30.15 mm) (Table 1). As shown in the M-A plot and volcano plot (Figure 2A,B), as the differentially expressed genes (DEGs) determined with a significance level of False Discovery Rate (FDR, <0.05) and an absolute fold-change (≥2), 80 marked up-regulated genes or 81 marked down-regulated genes were determined in these groups (the top 10 DEGs and all DEGs are shown in Table 2 and Appendix A). A hierarchical clustering heatmap (Figure 3) showed that gene expression profiles were (1) significantly different between the low AL and high AL groups, (2) almost identical within two independently prepared N1 samples (lowest AL) and (3) variable as AL increased.

To speculate the unidentified biofunctions related to the identified DEGs as above, GO enrichment analysis was employed. Interestingly, the top five significant canonical pathways were all related to cholesterol biosynthesis (Table 3) and the top five networks (score of more than 29, Table 4) were also related to (1) connective tissue disorders, hereditary disorder, organismal injury and abnormalities, (2) posttranslational modification, vitamin and mineral metabolism, auditory disease, (3) developmental disorder, neurological disease, (4) lipid metabolism, small molecule biochemistry, cellular movement and (5) cell morphology, nervous system development and function. Furthermore, to estimate what kinds of up-stream and/or network regulations induce such above mechanisms, Ingenuity Pathway Analysis (IPA, Qiagen, Redwood City, CA) was performed (Table 5). Sterol regulatory element-binding transcription factor 2 (SREBF2) was identified as both a possible upstream regulator and a causal network regulator. In addition, SREBF1, insulin-induced gene 1 (INSIG1) and insulin-like growth factor 1 (IGF1) were identified as upstream regulators, and protein tyrosine phosphatase receptor type O (PTPRO) was identified as a causal network regulator (Figure 4). Interestingly, among these, SREBP, INSIG and IGF1 were closely linked with each other (Figure 4A). Therefore, taken together, the results of the current study rationally suggested that HSSF cells having different biological activities may induce scleral elongation that results in myopic changes. 

## 3. Discussion

It is known that sterol regulatory element-binding proteins (SREBPs), comprising three major isoforms, SREBP1a, SREBP1c and SREBP2 [25,26], are pivotal membrane-bound transcription factors involved in the regulation of lipid biosynthetic gene expression [27]. Functionally, SREBP2 essentially activates genes involved in fatty acid (FA) and triglyceride (TG) synthesis, SREBP1c activates genes for cholesterol synthesis and SREBP1a can activate genes in both biosynthetic pathways [28,29]. In addition to these SREBPs, five additional proteins including the SREBP cleavage–activating protein (SCAP), INSIG1, INSIG2, Site-1 protease (S1P) and Site-2 protease (S2P) have been identified, and they constitute the core signaling pathway to activate SREBPs from their membrane-bound precursor forms (pSREBPs) to change into their active nuclear forms (nSREBPs) in response to sterol quantity [30,31,32,33,34]. In the current RNA sequencing analysis by GO ontology and IPA analyses, cholesterol biosynthesis-related canonical pathways were identified as the possible underlying responsible mechanisms causing myopic changes in HSSFs, and SREBP1, SREBP2, INSIG1, IGF1 and PTPRO may be upstream and/or causal master regulators. Among these possible regulators, SREBP1 and 2 form a functional complex with INSIG1 as stated above. In addition, IGF1 stimulates SREBP-1 expression induced by activation of the PI3-kinase/Akt signaling pathway [35], and PTPRO silencing can activate the Akt/mammalian target of rapamycin (mTOR) signaling axis, thereby promoting de novo lipogenesis by SREBP1 [36]. In terms of the relationship between SREBP2 and ocular pathogenesis, a recent study demonstrated that SREBP2-related defect in the cholesterol biosynthesis in eye lens cells may be related to cataract. That is, depletion of quaking (Qki) required for the activated expression of genes involved in biosynthesis of cholesterol resulting in a significant reduction of total levels of cholesterol in the ocular lens, eventually leading to cataracts. As possible mechanisms, it was also shown that Qki stimulates cholesterol biosynthesis via SREBP2 and Pol II, suggesting that Qki-SREBP2-related cholesterol biosynthesis could be pivotal for maintaining the cholesterol levels to avoid cataract development [37]. Alternatively, Graves’ ophthalmopathy (GO) is a disorder characterized as increased volume of the orbital fatty tissues caused by inflammation. As identified by GO etiology, stimulation by thyroid-stimulating hormone (TSH), IGF-1, IL-1, interferon γ and platelet-derived growth factor induce adipogenic differentiation of orbital fibroblasts (OFs) in the orbital fat as well as extraocular muscles [38]. A recent study demonstrated that phosphatase of regenerating liver-1 (PRL-1) overexpressed multilineage differentiation potential human placental mesenchymal stem cells (hPMSCsPRL-1) restore inflammation and adipogenesis of the GO model through the SREBP2/HMGCR (3-hydroxy-3-methyl-glutaryl-coenzyme a reductase) pathway [39]. Taken together, it is rationally speculated that SREBP-related signaling may play an essential role in the pathogenesis of the various ocular diseases including myopia, cataract, GO and others, and thus, in turn, could be used for potential therapeutic strategy for these ocular diseases.

IGF1 is well known to play a pivotal role in growth, development and metabolism [40]. Previous studies have shown that IGF1 contributes to eye growth as well as myopia development [41,42], and, in fact, the IGF-1 gene on chromosome 12q23.2 is located within the high-grade myopia locus, MYP3 interval, which has been mapped for autosomal dominant high myopia [43]. In fact, in addition to a number of SNPs within IGF-1 gene that have been reported in myopia [44], many recent genetic studies indicated that rs12423791 or rs6214 polymorphisms in IGF-1 were identified to be pivotally associated with high myopia in populations of Caucasian and Chinese people [45,46] but not in Japanese population [47,48]. Furthermore, a recent meta-analysis has demonstrated that the G allele of the IGF1 rs2162679 SNP is a potential protective factor for any myopia [49]. Taken together with the current observation by IPA that SREPB2 was commonly estimated as both the upstream regulator and causal network master regulator, possible lipid biosynthesis regulatory mechanisms related to a focus on SREBP2 may be involved in the unidentified mechanisms responsible for inducing myopic changes by HSSFs. Quite interestingly, a recent study demonstrated that SREBP2 in retinal pigment epithelium (RPE) cells may have a function to promote eye size in postnatal mice by repressing low-density lipoprotein receptor-related protein 2 (Lrp2), which has been identified to qualify the eye overgrowth [50]. 

Several studies have demonstrated an essential role of endoplasmic reticulum (ER) stress in the pathogenesis of nonalcoholic fatty liver disease (NAFLD). For instance, overexpression of glucose regulatory protein 78 (GRP78) in the livers of ob/ob mice reduced ER stress and inhibited SREBP-1c cleavage, inducing a decrease in hepatic triglyceride and cholesterol levels [51]. Alternatively, transcription factor 6α (ATF6α) or inositol-requiring enzyme 1α (IRE1α) is activated upon acute ER stress or upon the administration of a high-fat diet (HFD) [52,53]. Therefore, these observations indicated that SREBPs are significantly corelated with ER stress, and in our preceding study, mRNA expressions of various ER stress-related factors were in fact found to be substantially modulated among HSSF cells obtained in different AL groups [24]. It was also revealed that SREBP-1c is involved in caloric restriction (CR)-induced mitochondrial activation obtained by upregulating a master regulator of mitochondrial biogenesis, called peroxisome proliferator-activated receptor γ coactivator-1α (PGC-1α). Since CR-controlled phenotypes of tissues were recognized only within white adipose tissue (WAT), it was suggested that CR rationally causes SREBP-1c-related metabolic changes such as the elevation of the FA biosynthesis as well as activated mitochondrial functions via PGC-1α in WAT [54]. Considering the fact that numerous transcription factors and coactivators, including SREBP, PGC-1α and others, are involved in regulation of the expression of enzymes catalyzing the rate-limiting steps of liver glycolytic metabolic processes [55], it is reasonable to speculate that SREBP-related mechanisms may also induce alterations in both OCR and ECAR indices among HSSF cells obtained from eyes with various ALs, as shown in Figure 1. To support our current evidence, it is shown that high myopia may be induced by oxidative stress base upon family and population genetic studies identifying the nuclear-genome variants within proteins related to mitochondrial function rationally suggesting the critical roles of mitochondrial variants in the molecular etiology of high myopia [56].

In terms of possible systemic factors related to myopia pathogenesis, a recent two-sample Mendelian randomization (MR) study demonstrated that among six glycemic factors including adiponectin, body mass index, fasting blood glucose, fasting insulin, hemoglobin A1c (HbA1c) and proinsulin levels, adiponectin was negatively associated with myopia incidence, and a higher HbA1c level was associated with a greater risk of myopia, suggesting systemic glycemic regulation factors could be possible therapeutic targets for myopia onset [57]. It is well known that in type 1 diabetes, circulating IGF1 levels are negatively associated with glycemic regulation [58], and interestingly, previous non-targeted metabolomics analysis by ultra-HPLC with tandem mass spectrometry resulted that four metabolic pathways, insulin secretion, bile secretion, thyroid hormone synthesis and cGMP-PKG signaling pathway, were enriched in both aqueous humor and vitreous humor in patients with pathogenic myopia as compared with age- and sex-matched controls [59]. Further analysis using an insulin microarray resulted that the levels of insulin and those related factors including IGF2, IGF-2R, IGF binding protein 1 (IGFBP-1), IGFBP-2, IGFBP-3, IGFBP-4 and IGFBP-6 were markedly increased in patients with pathogenic myopia as compared with the controls [59]. Alternatively, the multivariable analysis using 2213 participants without retinal or optic nerve diseases underwent a series of measurements including spectral-domain optical coherence tomography, which indicated that a thicker retinal outer nuclear layer was associated (correlation coefficient r: 0.40) with shorter axial length and shorter disc–fovea distance after adjusting for younger age, male sex (beta: 0.24; *p* < 0.001), lower serum cholesterol concentration (beta: −0.05; *p* = 0.04) and thicker subfoveal choroidal thickness [60]. Therefore, this collective evidence rationally supported not only that systemic glycemic regulation factors were indeed involved in the pathogenesis of myopia but also the current investigation suggesting that SREBP-related regulation of FA, TG and cholesterol synthesis may be one of the critical factors to induce AL elongation via HSSFs in myopic eyes. 

However, as a limitation of the current study, we used only five elderly donors and did not perform an in vivo study. Since the production of ECMs by HSSFs are increasing until the globe reaches adult size at approximately 10 years of age [13], younger donors, especially below 10 years of age, would be ideal to confirm our current observations. Therefore, for a better understanding of SREBP-related molecular mechanisms in myopic etiology, additional in vivo studies using larger numbers of HSSF specimens obtained from various aged donors as well as SREBP knockout models will be required as our next project.

## 4. Materials and Methods

In the current study, we performed at Sapporo Medical University Hospital, Japan, after the all experimental procedures had been approved by the institutional review board (IRB registration #282-8). In addition, experiments using human-related specimens, human scleral stromal fibroblasts (HSSFs), were exclusively conducted according to the tenets of the Declaration of Helsinki and national laws for the protection of personal data including written informed consent obtained from all participants in this study. 

### 4.1. Preparation of HSSFs

We used surgically obtained scleral specimens obtained during a scleral shortening procedure of 5 patients with refractory rhegmatogenous retinal. The axial length (AL, mm) and ages (years-old) of these 5 patients are described in Table 1. Those were then subjected to isolation and preparation of HSSFs, as mentioned in our previous report [61]. In brief, these HSSFs cells were then processed to 2D planar culture using 150 mm 2D dishes until their confluency was reached at approximately 90% using HG-DMEM cell culture medium supplemented to 10% FBS, 2 mM L-glutamine, 100 units/mL penicillin G, 100 μg/mL Streptomycin Sulfate, 0.25 μg/mL amphotericin B under conditions at 37 °C and 5% CO_2_. For these, 2D cultures were maintained by exchange of the culture medium every other day before use. 

### 4.2. Measurements of Mitochondrial Respiration and Glycolytic Functions in 2D Cultured HSSF Cells

Two-dimensional cultured HSSF cells (20 × 10^3^) prepared as above were placed into a well of a plastic cell culture microplate that the manufacturer recommended for a Seahorse XFe96 Bioanalyzer (#103794-100, Agilent Technologies, Santa Clara, CA, USA) before the day of analysis, and the plates were cultured under conditions at 37 °C and 5% CO_2_. On the day of the analysis, the culture medium was changed into 180 μL of manufacturer-recommended DMEM assay medium (#103575-100, Agilent Technologies) containing 5.5 mM glucose, 2.0 mM glutamine and 1.0 mM sodium pyruvate. After the pre-incubation in a CO_2_-free incubator at 37 °C for 1 h, the assay plates were subjected to the protocol for measurements of mitochondrial (OCR; oxygen consumption rate) and glycolysis function (ECAR; extracellular acidification rate) by monitoring using a Seahorse XFe96 Bioanalyzer, as mentioned in our previous reports [62,63].

### 4.3. RNA Sequencing Analysis and Estimation of Possible Biological Functions

Total RNA extraction (n = 3 each) from 2D cultured HCSFs obtained from various AL eyes as described above using a commercially available RNeasy mini kit (Qiagen, Valencia, CA, USA) was processed according to the manufacturer’s protocol. After confirmation of suitable content and quality of the prepared RNA for the RNA sequencing analysis and quantitative real-time PCR (RIN; RNA integrity number, >8.5), RNA sequencing analysis was performed as described recently [64]. Then, adapter sequence, ambiguous nucleotides and low-quality sequences were removed, and those clean-up reads were mapped as according to the reference genome sequence (GRCh38) using HISAT2 tools software (HISAT 2.2.1, accessed on 1 November 2023) [65]. The read counts for the genes were evaluated by featureCounts (version 1.6.3) and subjected to statistical analysis using DESeq2 (version 1.24.0). Differentially expressed genes (DEG) were defined by using statistical significance by an empirical analysis, genes with fold-change ≥ 2.0 and false discovery rate adjusted *p*-value < 0.05 and *q* < 0.08. 

The specific gene functions, that is, biofunctions and the canonical pathways, were estimated using gene ontology (GO) enrichment analysis [66] and possible upstream transcriptional regulators for the estimated biofunctions and the canonical pathways using ingenuity pathway analysis (IPA, Qiagen, https://www.qiagenbioinformatics.com/products/ingenuity-pathway-analysis, accessed on 1 November 2023) [67,68]. 

### 4.4. Other Analytical Methods

For statistical analyses, Graph Pad Prism 8 Software (San Diego, CA, USA) was used, and the statistical difference between groups was defined using Student’s *t*-test for two-group comparison or two-ANOVA, followed by Tukey’s multiple comparison test. All data are expressed as arithmetical means ± standard error of the mean (SEM).

## Figures and Tables

**Figure 1 ijms-25-00501-f001:**
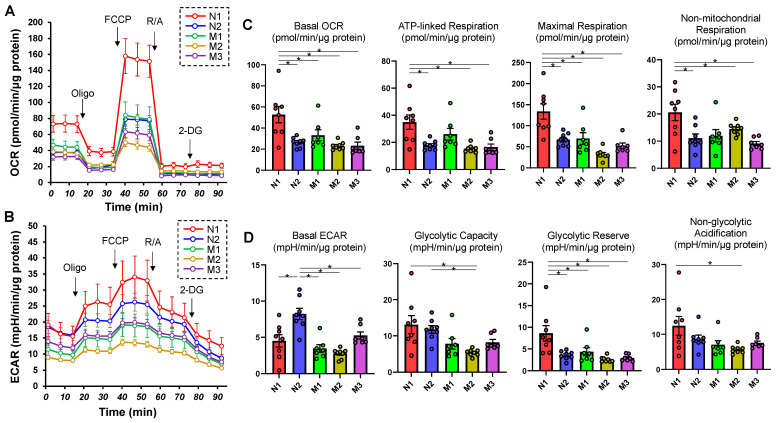
Measurements of mitochondrial and glycolytic functions in 2D HSSF cells. Real-time cellular biological functions of the 2D cultured HSSF cells were estimated using a Seahorse XFe96 analyzer. That is, the mitochondrial and glycolytic functions, oxygen consumption rate (OCR) and extracellular acidification rate (ECAR) were measured before treatment (at baseline) and after subsequent treatment with chemicals including oligomycin (Oligo: complex V inhibitor), FCCP (a protonphore), rotenone/antimycin A (R/A: complex I/III inhibitors) and 2-DG (hexokinase inhibitor). Fluctuations of the OCR and ECAR were plotted in (**A**,**B**), respectively. In addition, as the key parametric indices of OCR and ECAR, the mitochondrial respiration and glycolysis are shown in (**C**,**D**), respectively. * *p* < 0.05. Note that each sample is color-coded.

**Figure 2 ijms-25-00501-f002:**
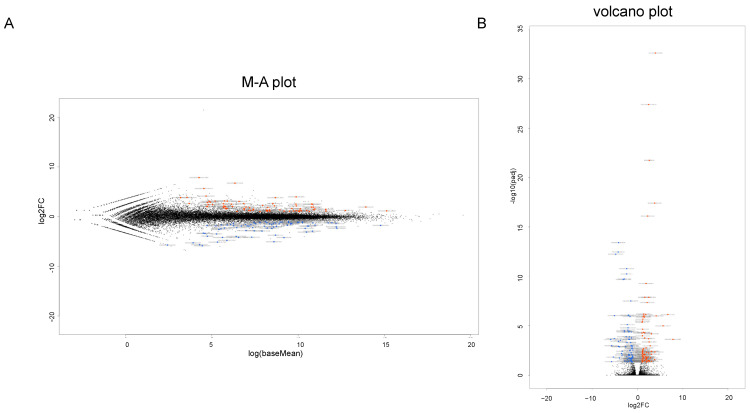
Differentially expressed genes (DEGs) in low AL and high AL groups. A total of 161 genes (80 up-regulated and 81 down-regulated genes) are identified as significant differentially expressed genes (DEGs) as determined by the following criteria: fold-change ≥ 2.0 and false discovery rate (FDR, adjusted *p*-value < 0.05 and q < 0.08) in the low AL and high AL groups. (**A**) The M-A plot representing the relationship between the mean expression values [log (baseMean); *x*-axis] and the magnitude of change in gene expression (log2 of fold-change; *y*-axis). (**B**) The volcano plots representing the relationship between the magnitude of gene expression change (log2 of fold-change; *x*-axis) and statistical significance of this change [−log10 of false discovery rate (FDR); *y*-axis]. Note that blue color represents low AL group and red color represents high AL group.

**Figure 3 ijms-25-00501-f003:**
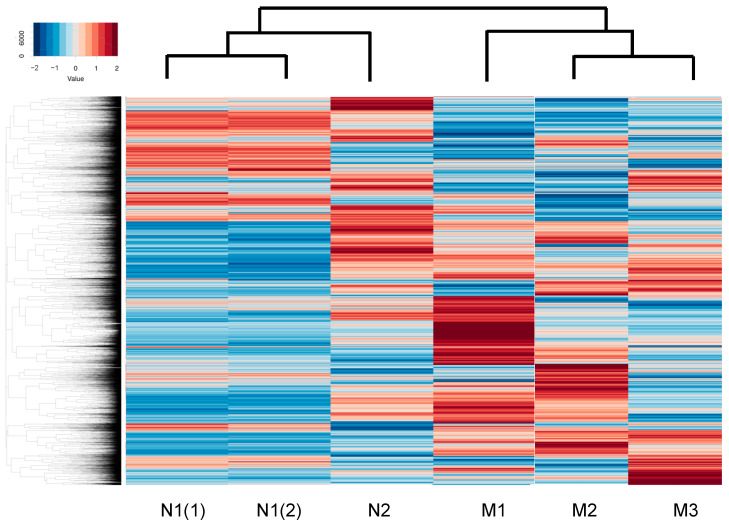
A hierarchical clustering heatmap.

**Figure 4 ijms-25-00501-f004:**
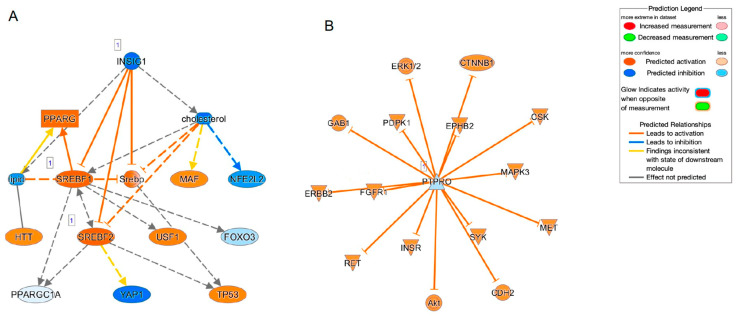
Causal network related to SREBF (**A**) and PTPRO (**B**).

**Table 1 ijms-25-00501-t001:** Baseline Characteristics of the subjects. N = normal sclera, M = myopic sclera.

HSSF	Axial Length (mm)	Age (Years Old), Gender
N1	22.8	69, female
N2	25.94	74, male
M1	27.5	69, female
M2	27.27	68, male
M3	30.15	67, female

**Table 2 ijms-25-00501-t002:** List of the top 10 up-regulated or down-regulated DEGs. Note that ↑ represents upregulation, whereas ↓ represents downregulation.

Expr Log Ratio ↑		Expr Log Ratio ↓	
Molecules	Value	Molecules	Value
HLA-DRB1	7.85	TGFA	−5.87
ROR2	6.72	MYH2	−5.72
TNFSF11	5.65	ZPLD1	−5.66
PRKG2	4.12	GAD1	−5.28
DMKN	3.96	SLITRK1	−5.13
COLCA2	3.82	HAS1	−5.06
SBSN	3.8	PTPRZ1	−4.78
SULT1C2	3.14	INA	−4.22
TNNT1	3.11	CCDC190	−4.18
EFNB3	3.08	EFHD1	−4.14

**Table 3 ijms-25-00501-t003:** Top canonical pathway.

Name	*p*-Value
superpathway of cholesterol biosynthesis	1.85 × 10^−13^
cholesterol biosynthesis I	1.61 × 10^−8^
cholesterol biosynthesis II (via 24,25-dihydrolanosterol)	1.61 × 10^−8^
cholesterol biosynthesis III (via desmosterol)	1.61 × 10^−8^
superpathway of geranylgeranyldiphosphate biosynthesis (via mevalonate)	5.62 × 10^−6^

**Table 4 ijms-25-00501-t004:** Top networks.

ID	Associated Network Functions	Score
1	Connective tissue disorders, hereditary disorder, organismal injury and abnormalities	42
2	post-translational modification, vitamin and mineral metabolism, auditory disease	40
3	developmental disorder, neurological disease, organismal injury and abnormalities	38
4	lipid metabolism, small molecule biochemistry, cellular movement	35
5	cell morphology, cellular movement, nervous system development and function	29

**Table 5 ijms-25-00501-t005:** Up-stream regulators and casual network regulators.

Up-Stream Regulator	Expr Fold Change	Molecular Type	Activation Z-Score
SREBF2	1.115	transcription regulator	3.352
SREBF1	1.268	transcription regulator	3.574
INSIG1	2.513	other	−3.724
IGF1	1.385	growth factor	1.041
Causal Network Regulator			
SREBF2	1.115	transcription regulator	3.402
PTPRO	−1.63	phosphatase	−0.271

## Data Availability

The datasets used and/or analyzed during the current study are available from the corresponding author on reasonable request.

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
