# Peer review of "Lipid Metabolism Regulators Are the Possible Determinant for Characteristics of Myopic Human Scleral Stroma Fibroblasts (HSSFs)"

_ijms, 2023, doi:10.3390/ijms25010501_

Round 1

Reviewer 1 Report

Comments and Suggestions for Authors

In this study, the authors showed a) alteration of gene expression depending on the axial length using human scleral cell culture by RNAseq and b) change of metabolism using an extracellular flux analysis. Both results strongly support metabolic association for myopia development and the shifts were maintained even after cell culture. Although the relationship between myopia and metabolism is not a new idea, this paper has precious results and is acceptable for publication. Before the publication, please address the major/minor points below. Also, please check English one more time.

Major points

1.       Table 4 was not mentioned in the main text.

2.       In many studies using IPA, SHEBFs were often selected as regulator genes.

https://pubmed.ncbi.nlm.nih.gov/31303399/

https://pubmed.ncbi.nlm.nih.gov/32705255/

https://pubmed.ncbi.nlm.nih.gov/35959888/

Of course, this gene may be so important, but I felt this is a pitfall of this analysis. Anyway, without subsequent experiments, the title may be too strong. I recommend changing the title, but no obligation.

Both results (Figure1 and RNAseq) indicated metabolic change. Although it is not a new idea, this is the fact found in this study.

3.       Please put the RNAseq data in a public database if possible. 

Minor points

line36, you may not need "It has been shown that". You can start with Myopia is the commonest....

line43, "the protective barrier to protect" may be "the barrier to protect".

line283, "1 % L-glutamine and 1 % Antibiotic-Antimycotic". Although I can understand as you diluted 100x solution to 1x, this is not the right description. Also, please mention the CO2 since you will use CO2 free incubation in the assay.

line290, "exchanged" should be "changed".

Supplemental table1. Please put the patient's gender. I think the axial length is an important parameter in this study. So, the supplemental table1 may be incorporated as a main table although the author mentioned them in line104-105.

line68,  HSSFs obtained from 7 patients. Why did you use only 5 HSSFs in this study? Were these 5 HSSFs different from the previous studies?

Figure1. Please put the sample number n in the legend.

line234, "is may", line337, “the is”.

The discussion was not interesting.

Comments on the Quality of English Language

Some of them are mentioned in "Comments and Suggestions", but not all. Please check the manuscript one more time.

Author Response

Major points

  1. Table 4 was not mentioned in the main text.

Answer; Thank you for this comment. As pointed out, Table 4 was mentioned in the last paragraph of result; “Furthermore, to estimate what kinds of up-stream and/or network regulations induce such above mechanisms, Ingenuity Pathway Analysis (IPA, Qiagen, Redwood City, CA) was performed (Table 4).”.

  1. In many studies using IPA, SHEBFs were often selected as regulator genes.

https://pubmed.ncbi.nlm.nih.gov/31303399/

https://pubmed.ncbi.nlm.nih.gov/32705255/

https://pubmed.ncbi.nlm.nih.gov/35959888/

Of course, this gene may be so important, but I felt this is a pitfall of this analysis. Anyway, without subsequent experiments, the title may be too strong. I recommend changing the title, but no obligation.

Both results (Figure1 and RNAseq) indicated metabolic change. Although it is not a new idea, this is the fact found in this study.

Answer; Thank you so much for this encouraging comment. As suggested, title was changed; “Lipid metabolism regulators are the possible determinant for characteristics of myopic human scleral stroma fibroblasts (HSSFs)”.

  1. Please put the RNAseq data in a public database if possible.

Answer; Thank you for this comment. As suggested, we would like to put the RNAseq data in a public database. Nevertheless, this may be difficult based upon contract with RNA seq analyzing company. However, if this is essential, we will ask company.

Minor points

  1. line36, you may not need "It has been shown that". You can start with Myopia is the commonest....

Answer; Thank you for this comment. As pointed out, this sentence was changed; “Myopia is the commonest refractive error leading to reversible visual impairment as well as blindness as the worst case worldwide [1-3].”.

  1. line43, "the protective barrier to protect" may be "the barrier to protect".

Answer; Thank you for this comment. As pointed out, this sentence was changed; "the barrier to protect".

  1. line283, "1 % L-glutamine and 1 % Antibiotic-Antimycotic". Although I can understand as you diluted 100x solution to 1x, this is not the right description. Also, please mention the CO2 since you will use CO2 free incubation in the assay.

Answer; Thank you for this comment. As pointed out, those culture conditions were more precisely described and included CO2 conditions; “HG-DMEM cell culture medium supplemented to 10 % FBS, 2 mM L-glutamine, 100 units/mL penicillin G, 100 mg/mL Streptomycin Sulfate, 0.25 mg/mL amphotericin B under conditions at 37 °C and 5 % CO2.”.

  1. line290, "exchanged" should be "changed".

Answer; Thank you for this comment. As pointed out, this sentence was changed; "changed”.

  1. Supplemental table1. Please put the patient's gender. I think the axial length is an important parameter in this study. So, the supplemental table1 may be incorporated as a main table although the author mentioned them in line104-105.

Answer; Thank you for this comment. As pointed out, the patient’s gender was included in the new Table 1 and place to line 104-105.

  1. line68, HSSFs obtained from 7 patients. Why did you use only 5 HSSFs in this study? Were these 5 HSSFs different from the previous studies?

Answer; Thank you for this comment. As pointed out, we would like use those HSSFs obtained from 7 patients. However, only 5 out of them were cultivated from their stock in the liquid nitrogen, and thus we used these 5 cells.

  1. Please put the sample number n in the legend.

Answer; Thank you for this comment. As pointed out, sample number n was included in the Fig. 1 legend; “A real time cellular biological functions of the 2D cultured HSSF cells were estimated to using a Seahorse XFe96 analyzer (n=8-10 each).”.

  1. line234, "is may", line337, “the is”.

Answer; Thank you for this comment. As pointed out, this sentence was changed; line234, "may be", line337, “that is”.

  1. The discussion was not interesting.

Answer; Thank you for this critical comment. As also suggested by another expert reviewer, study limitation in the Discussion was insufficiently described and thus, last paragraph was changed; “However, as a limitation of the current study, we used only five elderly donors and did not perform an in vivo study. Since the production of ECMs by HSSFs are increasing until the globe reaches adult size at approximately 10 years of age [13], younger donors especially below 10 years of age would be ideal to confirm our current observations. Therefore, for a better understanding of SREBP-related molecular mechanisms in myopic etiology, additional in vivo studies using larger numbers of HSSFs specimens obtained from various aged donors as well as SREBP knockout models will be required as our next project.”

Comments on the Quality of English Language

Some of them are mentioned in "Comments and Suggestions", but not all. Please check the manuscript one more time.

Answer; Thank you for this comment. As suggested, we carefully checked manuscript one more time.

Reviewer 2 Report

Comments and Suggestions for Authors

Immense work and lot of insight into a vague disease (myopia progression). Two remarks that need to be added

1-The manuscript is written in reverse with Materials and Methods after Discussion while it should be otherwise

2-Limitations should b mentioned:

A- samples are from elderly while below 10 would be ideal for the study as the authors mention that HSSF secrete ECK maximally till age 10

"Among the structural components of the human sclera, human scleral stroma fibroblasts (HSSFs) producing and secreting ECMs are known to be involved in pivotal roles during the development of the eye [7,10–12]. HSSFs increase the production of ECMs productions until the globe reaches adult size at approximately 10 years of age [13], and then the production of ECMs is decelerated and thereafter, the scleral ECM remodeling was maintained at physiological levels throughout the course of existence of an individual.."

B- surgical specimen carry the risk of being traumatic to the stromal fibroblasts by use of forceps and squeezing the specimen and are not ideal vs. clean harvesting. In case a long strip is removed the best yield would be to use the centre where forceps did not crush the tissue.

Author Response

Reviewer 2

Immense work and lot of insight into a vague disease (myopia progression). Two remarks that need to be added

  1. The manuscript is written in reverse with Materials and Methods after Discussion while it should be otherwise

Answer; Thank you for this comment. According to the IJMS submission guideline, Materials and Methods are placed after Discussion.

  1. Limitations should be mentioned:

A- samples are from elderly while below 10 would be ideal for the study as the authors mention that HSSF secrete ECK maximally till age 10

"Among the structural components of the human sclera, human scleral stroma fibroblasts (HSSFs) producing and secreting ECMs are known to be involved in pivotal roles during the development of the eye [7,10–12]. HSSFs increase the production of ECMs productions until the globe reaches adult size at approximately 10 years of age [13], and then the production of ECMs is decelerated and thereafter, the scleral ECM remodeling was maintained at physiological levels throughout the course of existence of an individual.."

B-surgical specimen carry the risk of being traumatic to the stromal fibroblasts by use of forceps and squeezing the specimen and are not ideal vs. clean harvesting. In case a long strip is removed the best yield would be to use the centre where forceps did not crush the tissue.

Answer; Thank you for those excellent comments. As suggested, I agree that comment A is extremely important and thus this issue is included; “However, as a limitation of the current study, we used only five elderly donors and did not perform an in vivo study. Since the production of ECMs by HSSFs are increasing until the globe reaches adult size at approximately 10 years of age [13], younger donors especially below 10 years of age would be ideal to confirm our current observations. Therefore, for a better understanding of SREBP-related molecular mechanisms in myopic etiology, additional in vivo studies using larger numbers of HSSFs specimens obtained from various aged donors as well as SREBP knockout models will be required as our next project. In terms of comment B, we also agree with those risk during the collection of the surgical specimens. However, we collected surgical scleral explant specimens were initially cultured and then collected human scleral stromal fibroblasts erupted around those explants. Therefore, since if scleral stroma is damaged, we can not obtain fibroblasts, our prepared HSSFs should be erupted from healthy stromal cells within the explant.
